# Squeeze-Type Piezoelectric Inkjet Printhead Actuating Waveform Design Method Based on Numerical Simulation and Experiment

**DOI:** 10.3390/mi13101695

**Published:** 2022-10-09

**Authors:** Ning Liu, Xianjun Sheng, Mingcong Zhang, Wei Han, Kexin Wang

**Affiliations:** 1School of Electrical Engineering, Dalian University of Technology, Dalian 116086, China; 2Office of Yuncheng Municipal Committee of CPC, Yuncheng 044000, China; 3School of Mechanical Engineering, Dalian University of Technology, Dalian 116086, China

**Keywords:** piezoelectric inkjet printhead, actuating waveform, numerical simulation, coupled multi-physics field, pressure variation

## Abstract

The piezoelectric inkjet printing technique has been commonly used to produce conductive graphics. In this paper, a trapezoidal waveform design method for squeeze-type piezoelectric inkjet printhead is presented to provide a modified steady ejection and optimal droplet shape, in which a coupled multi-physics model of a piezoelectric inkjet printhead is developed. This research describes the effects of parameters, including rising time *t_r_*, falling time *t_f_*, and dwelling time *t_d_*, of the trapezoidal waveform on the pressure at the nozzle through numerical simulations. These parameters are initially optimized based on numerical simulations and further optimized based on experimental results. When the printhead is actuated by the optimized waveform with the *t_r_* = 5 µs, *t_d_* = 10 µs, and *t_f_* = 2 µs, the droplets are in optimal shape, and their size is about half the diameter of the nozzle. The experimental results validate the efficacy of this waveform design method, which combines numerical simulation and experiment, as well as demonstrating that ink droplet formation can be studied from the point of pressure variation at the nozzle.

## 1. Introduction

Piezoelectric inkjet printing is a technology that uses piezoelectric inkjet print heads to deposit droplets on substrates and form graphics, and it has been widely used for printing patterns and documents [1]. Piezoelectric inkjet printheads not only have a strong control ability for droplets and high printing accuracy, but also need no heating, so they have more flexibility in ink compatibility [2]. Based on these features, many efforts have been made to use piezoelectric inkjet printing technology to directly deposit functional materials, thus expanding the application areas of piezoelectric inkjet printing technology. For example, piezoelectric inkjet printheads are used as manufacturing tools for printed electronics, and various types of conductive graphics are prepared using conductive inks containing effective particles [3]. This contributes to the fabrication of flexible electronics such as conformal antennas [4], wearable electronic devices [5,6], and biosensors [7].

With the expansion of piezoelectric printing applications, piezoelectric printing technology with higher accuracy is highly desired. For example, conformal antennas require ink droplet diameters down to the micron level [4]. Printing accuracy is affected by fluid characteristics, the print head structure, and even the actuating waveform [8]. During ejection, the piezoelectric inkjet printheads provide ink to the nozzle, while an actuating waveform is applied to the piezoelectric actuator to form a pressure wave. The pressure wave is propagated and reflected to generate droplets at the nozzle for ejection [9]. In the above process, the actuating waveform directly affects the velocity of the droplets, the ligament length, and the generation of satellite droplets [10]. Therefore, the actuating waveform can be optimized to improve the printing accuracy.

Based on Dijksman’s book, the trapezoidal waveform has strong controllability, large amplitude, and strong driving capability. This is also the reason why it was chosen as the actuating waveform of the printhead. At present, trapezoidal waves are widely used to actuate piezoelectric inkjet printheads. They include four parameters: rising time *t_r_*, falling time *t_f_*, dwelling time *t_d_*, and voltage amplitude *U* [11]. There have been several studies on the effect of trapezoidal waves on injection performance: the effect of the trapezoidal waveform on droplet volume and velocity in a bend-type piezoelectric inkjet printhead is discussed in [2]; the effect of the trapezoidal waveform on droplet morphology in squeeze-type piezoelectric inkjet printheads is investigated in [12]; and the effect of waveform parameters on droplet formation is discussed in [13] using numerical simulations and experiments. Therefore, the trapezoidal waveform needs to be properly designed to promote the piezoelectric inkjet printheads to generate droplets with optimal shape.

Most of earlier research focused on optimizing the *U* and *t_d_* of trapezoidal waveforms: a droplet-monitoring system is designed in [10] to record the droplet formation. It adjusts the *U* and *t_d_* in real-time so that the printhead can eject continuously. In [11], *t_d_* in the trapezoidal waveform is optimized by observing and measuring the motion of the meniscus at the nozzle to achieve high-speed ejection. This method enables the fast optimization of trapezoidal waveform parameters. Xiao et al. used Doppler vibrometry to determine *t_r_* and *t_f_* in the waveform and optimized *t_d_* by numerical simulation in [14]. This study demonstrates that this method can speed up the design efficiency of bend-type piezoelectric inkjet printhead structures.

The trapezoidal waveform design method for squeeze-type piezoelectric inkjet printheads is mainly optimized for the *U* and *t_d_*, ignoring the effects of *t_r_* and *t_f_* on the ejection. The *t_r_* and *t_f_* need to be precisely optimized by suitable methods to generate droplets with the optimal shape for a higher printing accuracy.

This paper proposes a waveform design method for squeeze-type piezoelectric inkjet printhead based on a numerical simulation and experiment. First, the coupled multi-physics field simulation software COMSOL Multiphysics is used for the preliminary design of the *t_r_*, *t_f_*, and *t_d_* of the trapezoidal waveform. Simulation results are compared to determine the optimal parameters and to analyze the effect of these parameters on pressure at the nozzle and droplet morphology. Then, a high-speed CCD camera with a microscope is used to observe the droplet formation to verify the simulation results. The parameters are further optimized according to the differences between the experiment and simulation. Actuated by the optimized waveform, the printhead can generate droplets with optimal shape. The droplet diameter, in this case, is about half the diameter of the nozzle, which is beneficial for improving printing accuracy.

## 2. Numerical Simulations

The commonly used piezoelectric inkjet printheads include squeeze-type, shear-type, bend-type, and push-type [15]. Squeeze-type single-nozzle printheads have a stronger jetting force than that of the commercially available multi-nozzle inkjet printhead. As a result, a wider range of ink can be jetted, and it is commonly used in ejection evaluation experiments to determine the optimal printing parameters [16]. Figure 1 shows the structure of the squeeze-type piezoelectric inkjet printhead (MJ-AT, Microfab, Plano, TX, USA) utilized in this study. The printhead length is 22.86 mm, the nozzle radius is 30 µm, and the taper angle at the nozzle is 15°. The printhead consists of a glass capillary tube with an actuated piezoelectric part adhered to the outside of the glass capillary tube [17]. The appropriate ink viscosity is an important guarantee of printing accuracy. The phenomenon of clogging can occur when viscosity is too high, while the contrary is a strong liquidity, resulting in unclear imaging. The test ink is nano-silver conductive ink (Jet-600C, HS Electronics) with a density of 0.95 g/cm^3^, viscosity of 6 mPa·s, and surface-tension coefficient of 27 mN/m. That satisfies the technical parameters of the piezo printhead. In an ejection, the actuating waveform is applied to the actuated piezoelectric part. The outer surface is connected to the positive electrode, and the inverse piezoelectric effect causes the piezoelectric part to expand and contract, eventually generating droplets.

The ejection of the piezoelectric inkjet printhead is numerically simulated using the commercial finite element analysis program COMSOL Multiphysics. As shown in Figure 2, the simulation model is coupled with a piezoelectric effect, fluid–structure interaction, and the two-phase flow (level set). The piezoelectric part is designed as a movable object. The fluid–structure interaction is bi-directional and real-time boundary data are transferred between the two physical fields. The moving mesh method is used to deal with the large-size deformation of solid structures and the construction of fluid–structure coupled interfaces. A fixed negative pressure boundary condition (800 Pa) at the ink supply channel is adopted to eliminate the effect of gravity on the ink. The numerical model uses adaptive mesh refinement to improve the accuracy of the simulation.

According to fluid dynamics theory, the incompressible N-S equation containing surface tension is used to describe the relationship between mass and momentum of the fluid.
(1)$$\frac{\partial \rho }{\partial t}+\Delta \cdot \rho v=0$$
(2)$$\rho \left(\frac{\partial v}{\partial t}+v\cdot \Delta v\right)=-\Delta p+\Delta \cdot \mu \left(\Delta v+{\left(\Delta v\right)}^{T}\right)+\rho g+F_{ST}$$
where *ρ* is the fluid density; ***v*** is the velocity vector of fluid flow; ∆ is the Hamiltonian operator; *p* is the pressure; *µ* is the fluid viscosity; and *F* is the volume force. The effect of surface tension on the injection process was not explicitly given in the text, but has now been supplemented in Chapter 2 as follows.

In order to obtain the accurate velocity of ink droplets, the effect of surface tension on ink droplet jetting must be considered, and the continuum surface force (CSF) model is used in this paper.
(3)$$F_{ST}=\sigma \delta \kappa n$$
where *σ* is the surface tension coefficient; *δ* is the interface Dirac delta function, which is non-zero only at the fluid interface; *κ* is the interface curvature; and ***n*** is the interface unit normal vector.

The *κ* and ***n*** can be expressed according to the level-set function as
(4)κ=−∇⋅n
(5)$$n=\frac{\nabla \phi }{\left|\nabla \phi \right|}$$

To simplify the calculation of surface tension, the *δ* function is approximated as
(6)δ=6|ϕ(1−ϕ)||∇ϕ|

There is a two-phase flow consisting of gas and liquid in the mode. The fluid interface between two immiscible fluids needs to be tracked in the numerical simulation. In this paper, the level-set method with reinitialization is used to track the two-phase flow interface, and the convective transport equation of this method is:(7)$$\frac{\partial \phi }{\partial t}+v\cdot \Delta \phi +\gamma \left(\left(\Delta \cdot \left(\phi \left(1-\phi \right)\frac{\Delta \phi }{\left|\Delta \phi \right|}\right)\right)-\varepsilon \Delta \cdot \Delta \phi \right)=0$$
where *ϕ* is the level-set variable; *γ* is the mobility; *ε* is the parameter-controlling interface thickness.

## 3. Preliminary Design of Trapezoidal Waveform Parameters

The finite element analysis software COMSOL is used to preliminarily optimize the trapezoidal waveform parameters considering the pressure variation at the nozzle. These parameters include rising time (*t_r_*), falling time (*t_f_*), and dwelling time (*t_d_*).

### 3.1. Optimization of t_r_ Based on the Pressure at the Nozzle and Meniscus Motion

The waveform shown in Figure 3 is designed to optimize the *t_r_*. Different voltage amplitudes will certainly affect the choice of waveform parameters [12]. However, it is confirmed that each piezoelectric nozzle has a range of applicable voltage–amplitude parameters. The voltage amplitude of 40 V was selected as the experimental parameter of the piezoelectric nozzle after referring to the manual of the piezoelectric nozzle. The *t_r_* and *t_d_* in the waveform are set to *t_r_* + *t_d_* = 50 µs. During the *t_r_*, the piezoelectric part expands, thereby creating a negative pressure in the chamber. This negative pressure causes the ink to be sucked back into the nozzle, while the ink is sucked into the chamber at the ink supply channel. Excessive ink suck-back is not conducive to droplet generation, as shown in Figure 4.

Hence, the negative pressure at the nozzle should be as small as possible to prevent excessive ink from being sucked into the nozzle. The *t_r_* is optimized by comparing the pressure variation at the nozzle at different *t_r_*, as shown in Figure 5. The shape of the pressure curve at the nozzle is similar at different *t_r_*. As the *t_r_* increases, the pressure variation at the nozzle and the maximum negative pressure decrease. According to the pressure wave propagation theory, the pressure wave generated during the *t_r_* is continuously propagated and reflected in the chamber [9]. Therefore, small droplets may be generated in the absence of positive pressure waves generated during the *t_f_*. If the *t_d_* is not set properly, the pressure waves generated during the *t_r_* and *t_f_* are propagated and reflected separately in the chamber, making it very easy to generate satellite droplets.

Therefore, the *t_r_* is optimized by comparing the simulation results of the meniscus motion at the nozzle at different *t_r_*, as shown in Figure 6. When the *t_r_* is 5 µs, the motion of the meniscus at the nozzle is so small that the printhead cannot easily generate satellite droplets. In summary, the optimal *t_r_* value is initially determined to be 5 µs.

### 3.2. Optimization of t_f_ Based on the Pressure at the Nozzle

The *t_f_* is optimized using a trapezoidal wave with a large value of *t_d_* as shown in Figure 7. The effect of the large value of *t_d_* is to make the pressure wave generated during the *t_r_* propagate and reflect continuously in the chamber. The falling edge of the voltage is applied to the printhead after the decay of this pressure wave. *U* is set to 40 V. To minimize the effect of the *t_r_* on the pressure at the nozzle, the *t_r_* is set to 5 µs according to the simulation results above. The *t_d_* is set to 95 µs, which means that the falling edge of the voltage is applied to the printhead at 100 µs. During the *t_f_*, the piezoelectric part contracts, thereby creating a positive pressure in the chamber. As the positive pressure increases, the ink is squeezed out of the chamber to form a ligament, as shown in Figure 8.

Ideally, this positive pressure is superimposed with the reflected positive pressure of the pressure wave generated during the *t_r_*, causing the positive pressure at the nozzle to increase. The droplets are squeezed out of the chamber under this positive pressure. To eject droplets, the positive pressure at the nozzle needs to be as high as possible. The *t_f_* is optimized by comparing the pressure variation at the nozzle at different *t_f_*, as shown in Figure 9. The shape of the pressure curve at the nozzle is similar at different *t_f_*. As the *t_f_* increases, the pressure variation at the nozzle and the maximum positive pressure decrease. Hence, the optimal *t_f_* value is initially determined to be 1 µs in order to generate a large positive pressure.

### 3.3. Optimization of t_d_ Based on the Pressure at the Nozzle and Droplet Formation

The *t_d_* is optimized using a trapezoidal waveform with different td, as shown in Figure 10. The waveform parameters are set as follows: *t_r_* = 5 µs, *t_f_* = 1 µs, *U* = 40 V. According to the pressure wave propagation theory, the optimal *t_d_* is to make the positive pressure generated during the *t_f_*, and the positive pressure obtained by reflection superimposed in the middle of the chamber generates droplets with an optimal shape. For this purpose, a sufficient pressure variation at the nozzle is required. The *t_d_* is optimized by comparing the pressure variation and velocity variation at the nozzle at different *t_d_*, as shown in Figure 11 and Figure 12. The shape of the pressure curve at the nozzle is distinctly different at different *t_d_*. When the td is 5 µs, both the maximum positive pressure and the maximum negative pressure are large, resulting in a large pressure variation. When the *t_d_* is in the range of 10~20 µs, the maximum positive pressure and the maximum negative pressure are appropriate, making the pressure variation moderate. When the *t_d_* is longer than 20 µs, the maximum positive pressure and the maximum negative pressure are small, so the pressure variation is small. While the same situation occurs in the velocity curve, moderate velocity occurs when *t_d_* is 10 µs.

The *t_d_* is further optimized based on the simulation results of droplet generation at the nozzle, as shown in Figure 13. When the *t_d_* is short, the droplets are generated with a long droplet tail due to excessive pressure variation. In this case, the printhead is likely to generate satellite droplets, which is not conducive to improving printing accuracy. When the *t_d_* is long, the printhead cannot squeeze the ink out of the nozzle because the positive pressure is too small. It is also possible that the negative pressure is too small, so that the ligament cannot break up. this case, even if droplets can be generated, they are too small to be used for printing.

As shown in Figure 13, when the *t_d_* is 10 µs, droplets with optimal shape are generated. The optimal *t_d_* value is initially determined as 10 µs to generate droplets with optimal shape. As the positive pressure increases, the ink is squeezed out of the chamber to form a ligament. Then, the positive pressure decreases, the negative pressure increases, and the ligament breaks up and forms a droplet with a round head and a slender tail. In order to verify the voltage applicability range of this parameter, the optimal actuating waveform at different driving voltages is simulated, and the results are shown in Figure 14. When the driving voltage is too small, the surface tension cannot be overcome, but the droplets show long trailing with satellite droplets with a large amplitude. The larger the amplitude of the voltage, the higher the speed of the liquid drop. Ultimately, when the voltage is greater than 30 V and less than 45 V, the jetting effect will not be unsatisfactory. The velocity contour of the optimal parameters is shown in Figure 15, which describes the velocity field change of the formation of the droplet. A significant increase in velocity is observed when droplets are formed.

In summary, the optimal parameters of the trapezoidal waveform initially determined by simulation analysis are *t_r_* = 5 µs, *t_d_* = 10 µs, and *t_f_* = 1 µs, and this applies to voltage amplitude greater than 30 V and less than 45 V. In this study, the numerical model uses adaptive mesh refinement instead of overall fine dissection. The sparsity of the mesh dissection affects the correctness of the numerical simulation results. In practice, it is difficult to generate the ideal waveform for the simulation with the existing equipment. Therefore, the simulation results need to be verified experimentally, and the waveform parameters need to be further optimized accordingly.

## 4. Experimental Research

The experimental setup for observing the morphology of droplets is shown in Figure 16. The observation was achieved by a high-speed CCD camera with a microscope (Photron FASTCAM SA-5, Photron Corporation, Tokyo, Japan). The camera worked with a backlit light source to take pictures. A combination of an arbitrary waveform signal generator (AFG3022, Tektronix, Beaverton, OR, USA) and a voltage amplifier (PZD700, TREK, Waterloo, WI, USA) was used to actuate the piezoelectric printhead (MJ-AT, Microfab, Plano, TX, USA). The control program was written on LabVIEW (LabVIEW 2017, National Instruments, Austin, TX, USA) to programmatically control the arbitrary waveform signal generator and high-speed CCD camera via a PC. After the PC generates a trigger signal, several shutter signals are used to control the camera to take pictures of the droplet formation, thus recording the droplet shape.

To verify the numerical simulated optimized parameters of the trapezoidal waveform, two sets of experiments were carried out.

In the first set, the effect of different tr on the droplet morphology is observed. The *t_d_*, *t_f_*, and *U* are set according to the optimized parameters determined by simulation (*t_d_* = 10 µs, *t_f_* = 1 µs, *U* = 40 V), and the actuating frequency is set to 1 kHz. Meanwhile, the *t_r_* is set to 1 µs and 10 µs, respectively. The experimental results are shown in Figure 17. 

When the *t_r_* is 1 µs, fluctuation in the meniscus at the nozzle occurs during 20 to 50 µs. According to the analysis of the numerical simulation, when the *t_r_* is short, negative pressure at the nozzle is large, resulting in excessive ink being sucked into the nozzle. When the positive pressure wave reaches the nozzle, it causes fluctuations in the meniscus, so no droplet is generated. When the *t_r_* is 10 µs, droplets with a long droplet tail are generated at the nozzle. This is because when the *t_r_* is long, negative pressure at the nozzle is small, resulting in a small amount of ink being sucked into the nozzle. When the positive pressure wave reaches the nozzle, it is easy for the printhead to generate droplets, thus forming a long droplet tail, which is not good for controlling the shape and size of the droplets.

In the second set, the effect of different *t_f_* on droplet morphology is observed. The *t_r_*, *t_d_*, and *U* are set according to the optimized parameters determined by simulation (*t_r_* = 5 µs, *t_d_* = 10 µs, *U* = 40 V), and the actuating frequency is 1 kHz, setting *t_f_* to 1 µs and 5 µs, respectively. The experimental results are shown in Figure 18.

When the *t_f_* is 1 µs (the optimized parameters determined by simulation), the ink is squeezed out of the chamber to form a ligament. The ligament then breaks up and forms a droplet with a round head and a slender tail. According to the analysis of the numerical simulation, the appropriate pressure variation allows the printhead to generate droplets that are close to the ideal shape. When the *t_f_* is 5 µs, fluctuation in the meniscus at the nozzle occurs during 20 µs to 50 µs. This is because the positive pressure at the nozzle is too small to squeeze the ink out of the nozzle.

When the printhead is actuated by the optimized waveform with the *t_r_* = 5 µs, *t_d_* = 10 µs, and *t_f_* = 1 µs, droplets with a slender droplet tail are generated at the nozzle. Therefore, further optimization of the parameters is required to generate droplets with optimal shape. According to the pressure wave propagation theory and the results of the numerical simulation, when the *t_f_* is 1 µs, the positive pressure at the nozzle is large, causing the printhead to squeeze out too much ink. This results in a droplet with a long droplet tail, which is not conducive to improving printing accuracy. Therefore, to reduce the positive pressure during ejection, the *t_f_* is set to 2 µs (other parameters are the same as in the second set) for the third set. The experimental result is shown in Figure 19, where the nozzle continuously ejects droplets with the optimal shape. The image processing and measurement program is written on LabVIEW using the Vision Development Module. The front panel of the program is shown in Figure 20. The diameter of this droplet is 29.2134 µm, which is about half the diameter of the nozzle. The droplet size is optimal, and the diameter and speed can be changed by varying the voltage amplitude.

In summary, when combining numerical simulations and experiments, the optimal waveform parameters are *t_r_* = 5 µs, *t_d_* = 10 µs, *t_f_* = 2 µs, *U* = 40 V. The actuating frequency is 1 kHz. Figure 19 shows the sequential images of the droplet formation: (1) the piezoelectric part expands, creating a negative pressure in the chamber, and the printhead squeezes the ink out of the nozzle at 20 µs; (2) the ink is squeezed out of the chamber to form a ligament at 40 µs; (3) the ligament length increases with time during 40 to 50 µs; (4) the ligament breaks up at 60 µs; and (5) during 60 to 80 µs, the squeezed ink forms a droplet due to surface tension.

## 5. Conclusions

The proper actuating waveform can considerably improve the ejection performance of a piezoelectric printhead. This paper proposes a waveform design method for squeeze-type piezoelectric printheads based on a numerical simulation and experiment for the printing accuracy improvement of ejecting nano-silver conductive ink. A coupled multi-physics field piezoelectric printhead ejection model is developed using the commercial finite element analysis software COMSOL to initially optimize the parameters from the point of view of the effect on the pressure at the nozzle. To prevent excessive ink suck-back and satellite droplets, the optimal tr is initially determined to be 5 µs. To produce a large positive pressure at the time of the ejection, the optimal *t_f_* is initially determined to be 2 µs. The generation of droplets requires an appropriate pressure variation at the nozzle, so the optimal *t_d_* is initially determined to be 10 µs. After that, experiments are conducted with other parameters fixed, and only *t_r_* and *t_f_* are changed. The experimental phenomena verify the simulation results, and the parameters are optimized further based on the experiments. Considering both the numerical simulations and experimental results, the optimal parameters of the actuating waveform are *t_r_* = 5 µs, *t_d_* = 10 µs, *t_f_* = 2 µs, and *U* = 40 V. The results show that the squeeze-type piezoelectric printhead can eject continuously when actuated by the designed waveform. The droplets are in optimal shape, and the size is about half the diameter of the nozzle. Meanwhile, the method also demonstrates the feasibility of analyzing the droplet formation from the perspective of pressure variation at the nozzle.

## Figures and Tables

**Figure 1 micromachines-13-01695-f001:**
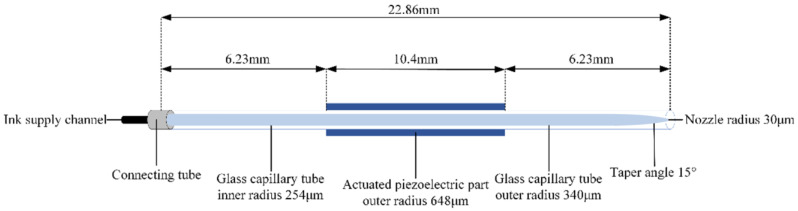
Schematic of squeeze-type piezoelectric inkjet printhead.

**Figure 2 micromachines-13-01695-f002:**
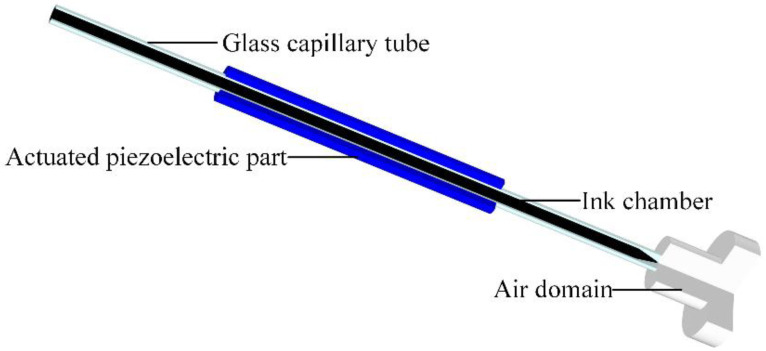
Numerical simulation model built in COMSOL Multiphysics.

**Figure 3 micromachines-13-01695-f003:**
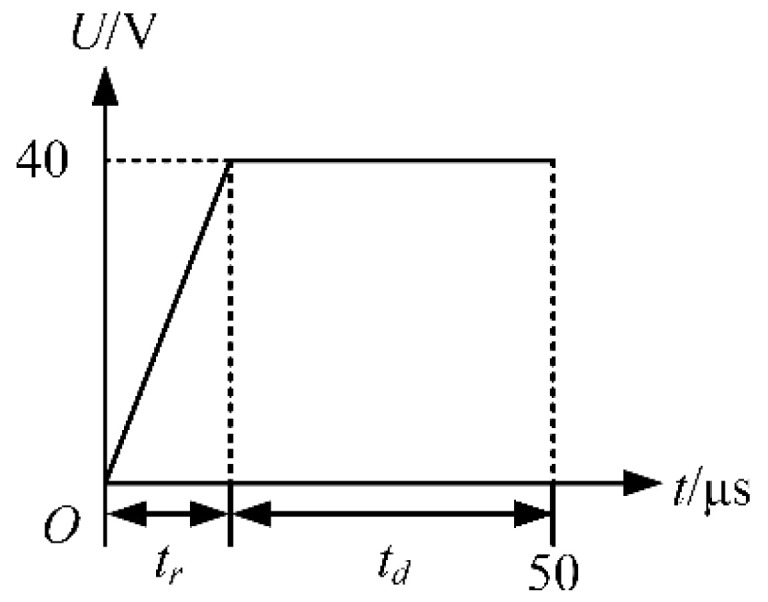
Waveform used in the optimization of *t_r_*.

**Figure 4 micromachines-13-01695-f004:**
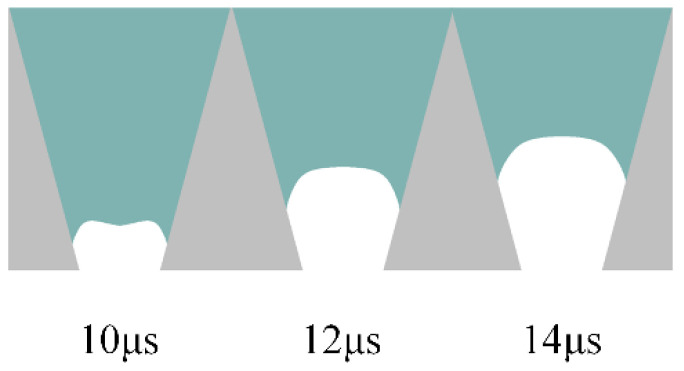
Excessive ink is sucked into the nozzle when the *t_r_* is 1 µs.

**Figure 5 micromachines-13-01695-f005:**
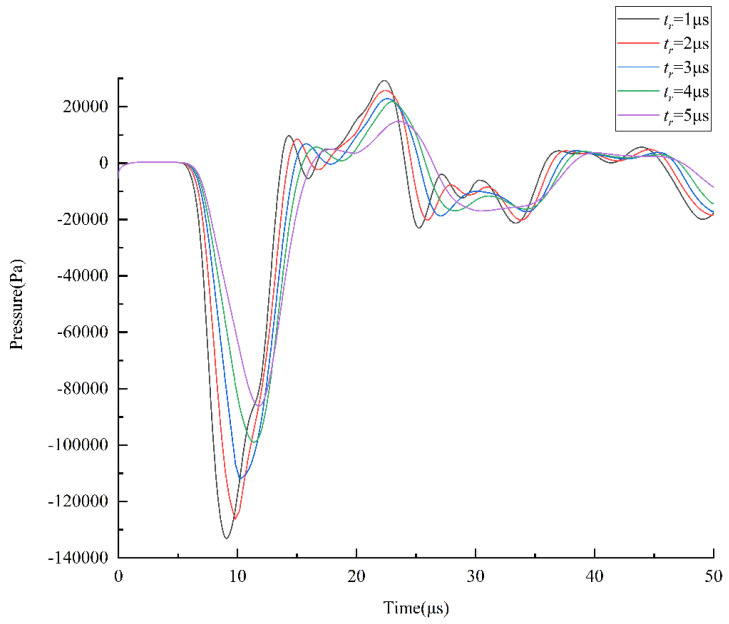
Effect of *t_r_* on the pressure at the nozzle.

**Figure 6 micromachines-13-01695-f006:**
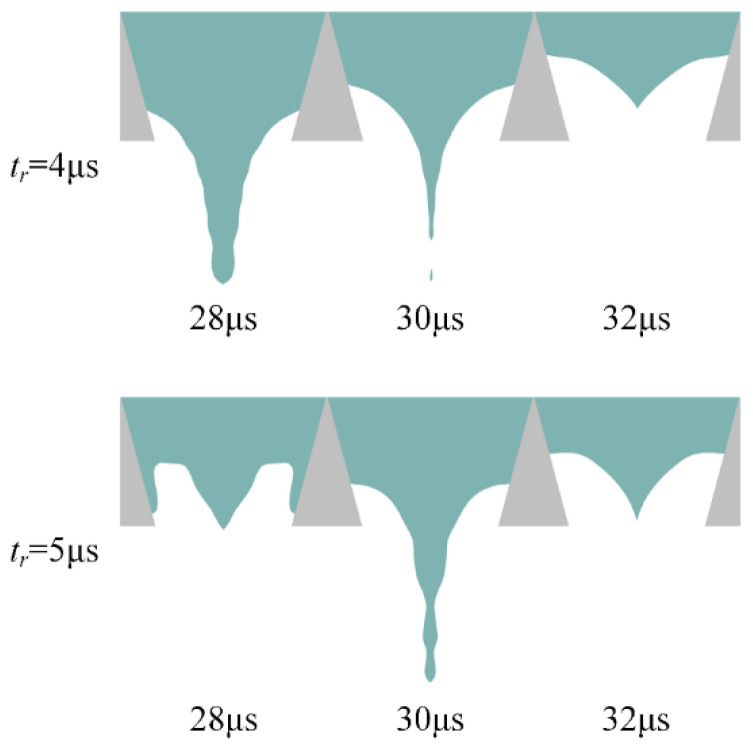
Simulation results of meniscus motion at the nozzle when the *t_r_* is 4 µs and 5 µs.

**Figure 7 micromachines-13-01695-f007:**
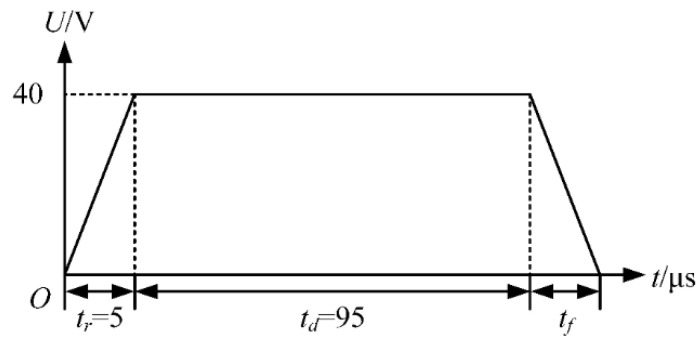
Trapezoidal waveform with a large value of *t_d_*.

**Figure 8 micromachines-13-01695-f008:**
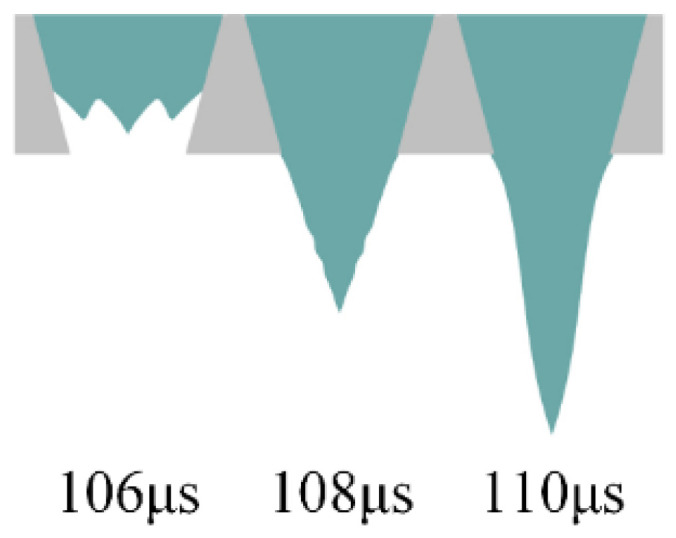
Ink is squeezed out of the chamber to form a ligament when the *t_f_* is 1 µs.

**Figure 9 micromachines-13-01695-f009:**
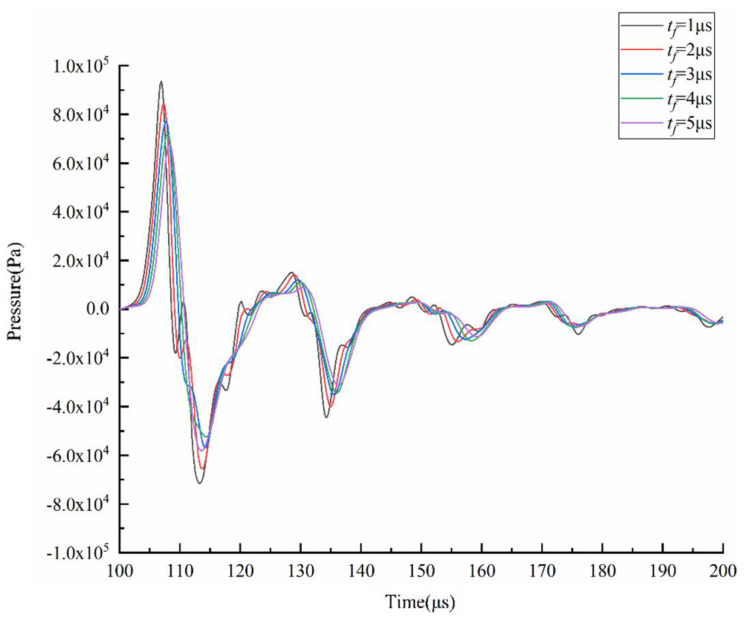
Effect of *t_f_* on the pressure at the nozzle.

**Figure 10 micromachines-13-01695-f010:**
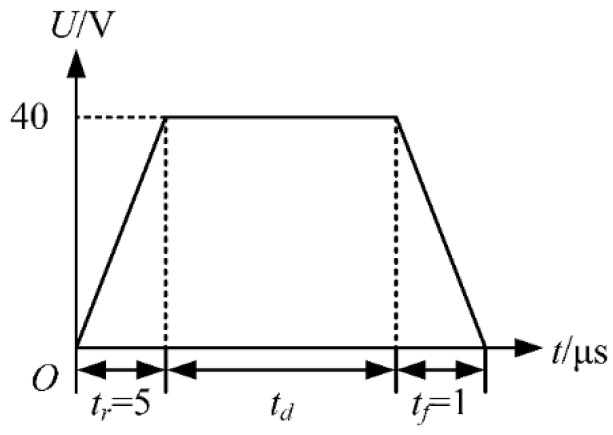
Trapezoidal waveform.

**Figure 11 micromachines-13-01695-f011:**
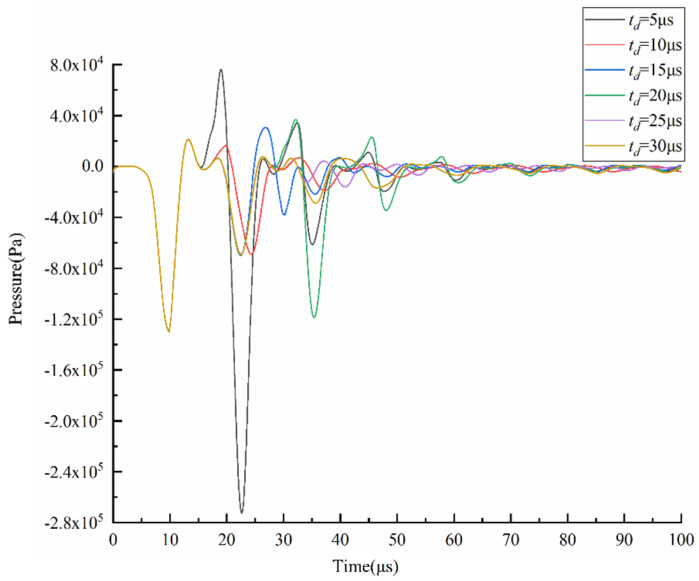
Effect of *t_d_* on the pressure at the nozzle.

**Figure 12 micromachines-13-01695-f012:**
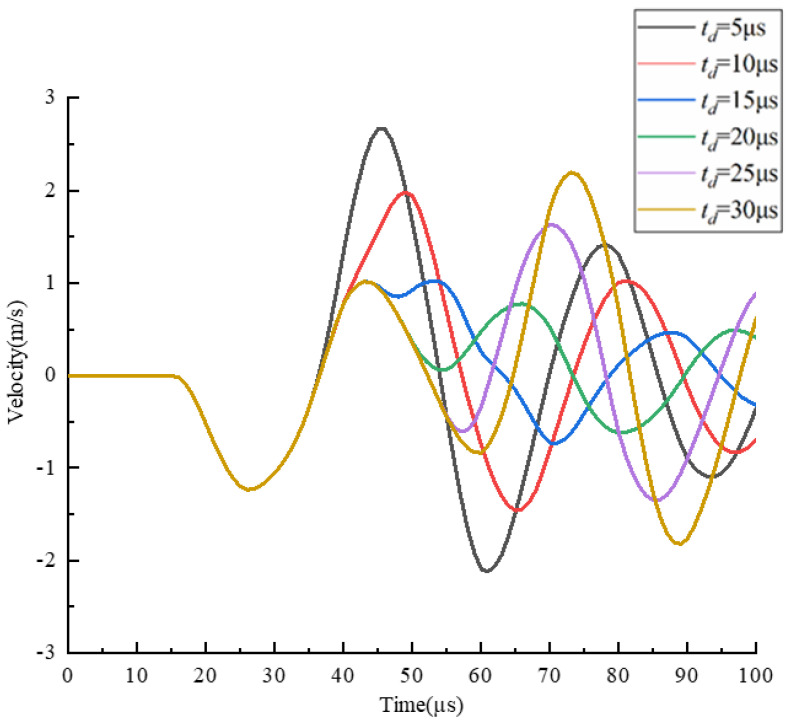
Effect of *t_d_* on the velocity at the nozzle.

**Figure 13 micromachines-13-01695-f013:**
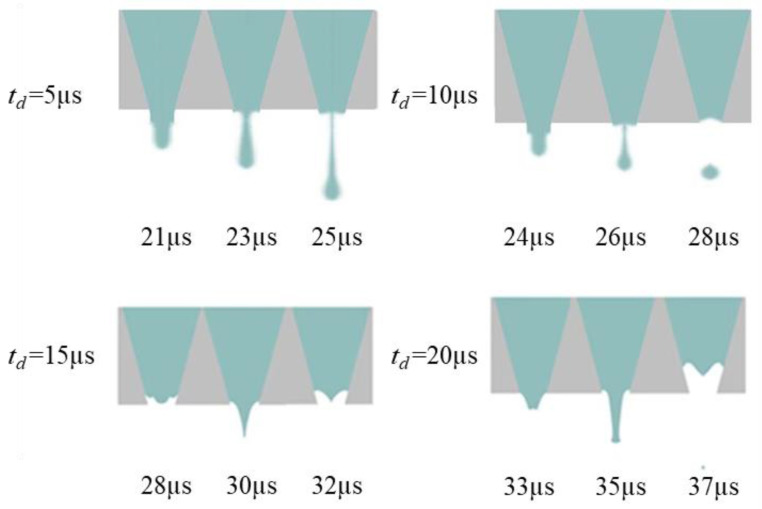
Simulation results of droplet formation at the nozzle at different *t_d_*.

**Figure 14 micromachines-13-01695-f014:**
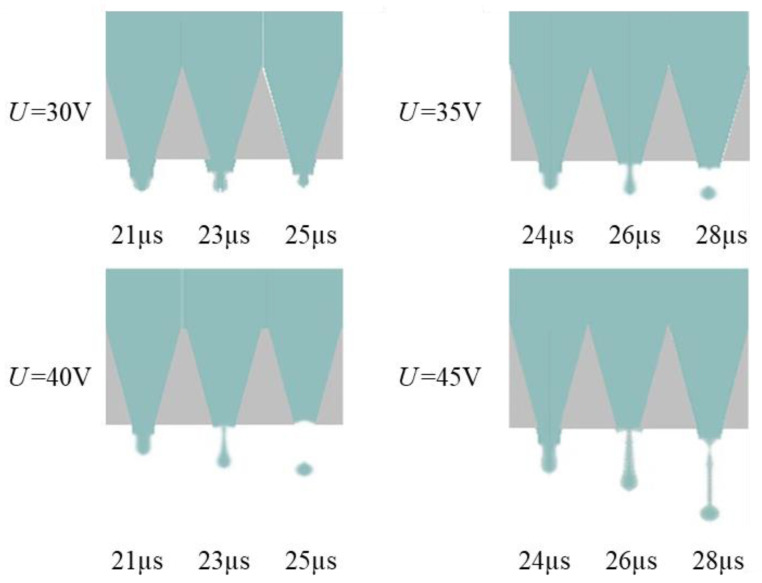
Simulation results of droplet formation at the nozzle at different *U*.

**Figure 15 micromachines-13-01695-f015:**
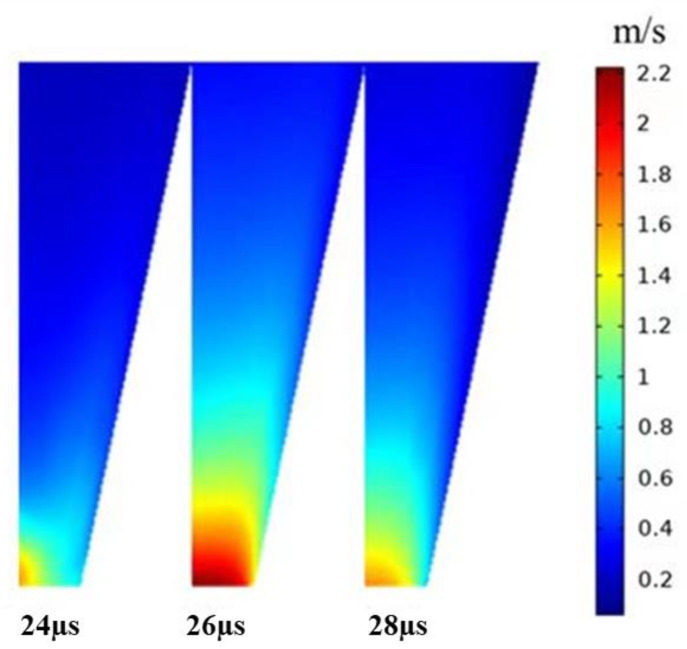
The velocity contour of the optimal parameters.

**Figure 16 micromachines-13-01695-f016:**
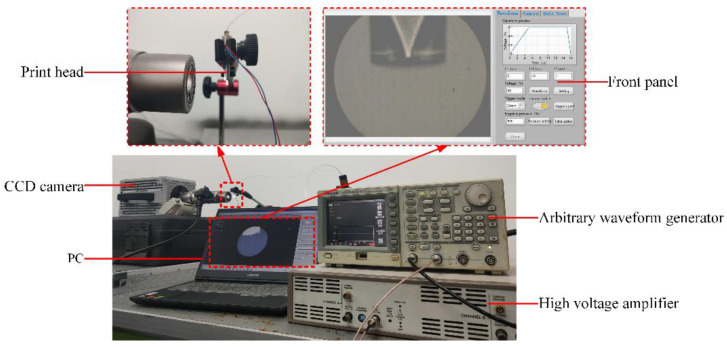
Experimental setup.

**Figure 17 micromachines-13-01695-f017:**
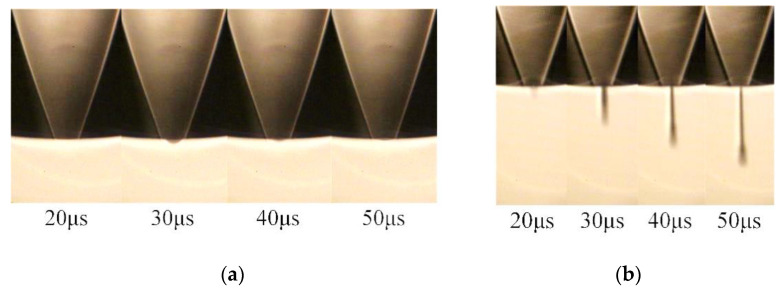
Results of the first set: (**a**) the *t_r_* is 1 µs (**b**) the *t_r_* is 10 µs.

**Figure 18 micromachines-13-01695-f018:**
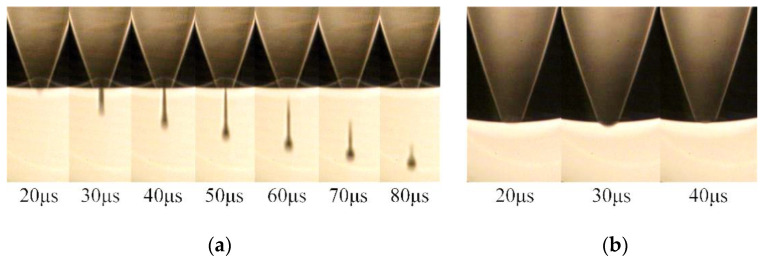
Results of the second set: (**a**) the *t_f_* is 1 µs (**b**) the *t_f_* is 5 µs.

**Figure 19 micromachines-13-01695-f019:**
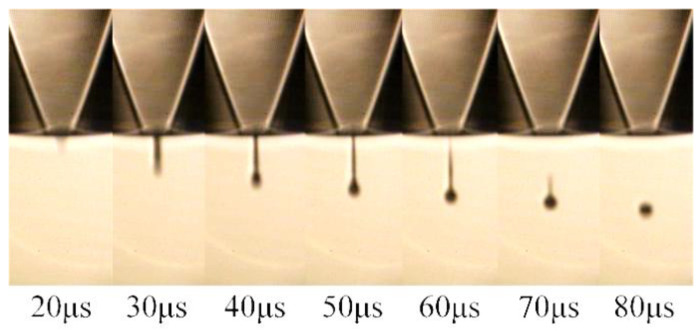
Experimental results when the *t_f_* is 2 µs.

**Figure 20 micromachines-13-01695-f020:**
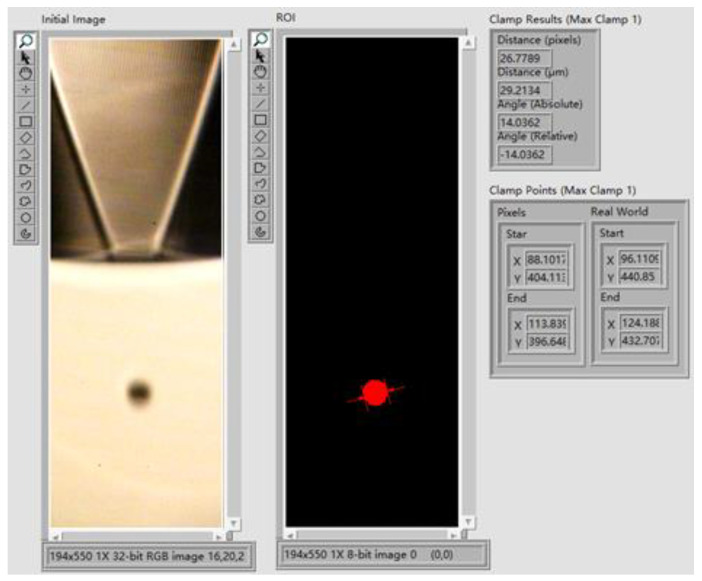
The front panel of the image processing and measurement program.

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
