# Peer review of "Squeeze-Type Piezoelectric Inkjet Printhead Actuating Waveform Design Method Based on Numerical Simulation and Experiment"

_micromachines, 2022, doi:10.3390/mi13101695_

Round 1

Reviewer 1 Report

Fig 1:  taper angle only approximates the inner shape of the MicroFab nozzles at exit, which are better described using hyperbolic tangents in my experience.

line 121: what do you mean by a reinitialization parameter?

line 157: the "optimal value of 5us" is really that for which meniscus barely moves for U=40V. Why not show this is still true for other choices of U?

line 194: better English "chamber generate droplets" as "to" makes no sense.

Figure 11: shows negative pressure exceeding 1bar, i.e. cavitation! Comment

Figure 12: with td-10us but drop volumes at 26us and 28us clearly differ. This means there is no volume conservation somewhere - you blame meshing? If that is true why use these simulations to guide your method?

Figure 13: Sideways jetting shown experimentally, but downwards images! No comment about this set up appears in the text.

Figure 14: shows poor alignment of jetting axis with the camera image plane and such a poor contrast between air and ink within the tapered nozzle that I nearly missed the extreme drawback in 14 (a) that you described well. Please attempt to improve this for presentation, even if artificial, as I have achieved far better using such Microfab nozzles, despite the challenges!

Comparing Figure 15(a) with Figure 16 shows a considerable reduction in both size and speed of the jetted drop, and yet you make no mention of the latter, which can be important for accurate drop location and spreading on impact wit the substrate.. This also means that the "optimum" U=40V with your optimised waveform would be insufficient if you required higher speed.

You make no reference to the work of J. Frits Dijksman, who analytically described tubular inkjet systems in many papers from 1984 onwards to his latest book published in 2019. You should at least have a look to discover you could have replaced your inkjet simulations with a good approximation. 

Hydrodynamics of small tubular pumps, J. Fluid Mechanics , 139  ,1984) , pp. 173 - 19

  Inkjet Technology for Digital Fabrication,Wiley 2012 Ian M. Hutchings,andGraham D. Martin (Eds)Chapter 3 Dynamics of Piezoelectric Print-Heads, by J. Frits Dijksman and  Anke Pierik

J. Frits Dijksman, Design of Piezo Inkjet Print Heads: From Acoustics to Applications, Wiley (2019)

Reviewer 2 Report

Thank you for the interesting work.

Questions:

1. Could you write some suggestions about how it will affect the result / print quality if the ink has a different viscosity?

2. If possible, could you also show a figure of the ink inlet velocity at the nozzle, taking into account various t_d?

Figure 11 shows t_d as t_dwell.

3. Equation 2. 

Could you write why you don't take into account the acceleration due to gravity?

Also, you mentioned "F is the volume force ", where is it?

4. How do you define surface tension force and how do you consider it for ink movement?

Reviewer 3 Report

In this paper, the authors study the impact of different waveform parameters to implement optimal droplet size during piezoelectric inkjet printing. In particular, they study the impact of the rise time, dwell time and fall time on the pressure of the nozzle and generated droplets using COMSOL simulations, and establish that a rise time of 5 us, dwell time of 10 us, and fall time of 2 us leads to optimal droplet generation. They also validate these results experimentally.

This work is carefully performed and the experiments and discussion are well explained. This study does a good job in helping practitioners understand the impact of different parameters. I recommend the publication of this manuscript. 

It would be great if the authors could incorporate the minor revisions below to further improve the quality of the work:

- Is there any work that establishes a trapezoidal waveform as the best one for printing? Is there any literature on other waveforms? If so please include these references in the introduction.

- Please include the pressure field near the nozzle as a heat map at different time points (for example accompanying Fig 6,8,12) so it complements the discussion around droplet generation mechanisms.
